# Correlation Priors for Reinforcement Learning

**Bastian Alt**[*]    **Adrian Šošić**[*]    **Heinz Koeppl**
Department of Electrical Engineering and Information Technology
Technische Universität Darmstadt
{bastian.alt, adrian.sosic, heinz.koeppl}@bcs.tu-darmstadt.de

## Abstract

Many decision-making problems naturally exhibit pronounced structures inherited from the characteristics of the underlying environment. In a Markov decision process model, for example, two distinct states can have inherently related semantics or encode resembling physical state configurations. This often implies locally correlated transition dynamics among the states. In order to complete a certain task in such environments, the operating agent usually needs to execute a series of temporally and spatially correlated actions. Though there exists a variety of approaches to capture these correlations in continuous state-action domains, a principled solution for discrete environments is missing. In this work, we present a Bayesian learning framework based on Pólya-Gamma augmentation that enables an analogous reasoning in such cases. We demonstrate the framework on a number of common decision-making related problems, such as imitation learning, subgoal extraction, system identification and Bayesian reinforcement learning. By explicitly modeling the underlying correlation structures of these problems, the proposed approach yields superior predictive performance compared to correlation-agnostic models, even when trained on data sets that are an order of magnitude smaller in size.

## 1   Introduction

Correlations arise naturally in many aspects of decision-making. The reason for this phenomenon is that decision-making problems often exhibit pronounced *structures*, which substantially influence the strategies of an agent. Examples of correlations are even found in stateless decision-making problems, such as multi-armed bandits, where prominent patterns in the reward mechanisms of different arms can translate into correlated action choices of the operating agent [7, 9]. However, these statistical relationships become more pronounced in the case of contextual bandits, where effective decision-making strategies not only exhibit temporal correlation but also take into account the state context at each time point, introducing a second source of correlation [12].

In more general decision-making models, such as Markov decision processes (MDPs), the agent can directly affect the state of the environment through its action choices. The effects caused by these actions often share common patterns between different states of the process, e.g., because the states have inherently related semantics or encode similar physical state configurations of the underlying system. Examples of this general principle are omnipresent in all disciplines and range from robotics, where similar actuator outputs result in similar kinematic responses for similar states of the robot's joints, to networking applications, where the servicing of a particular queue affects the surrounding network state (Section 4.3.3). The common consequence is that the structures of the environment are usually reflected in the decisions of the operating agent, who needs to execute a series of temporally and spatially correlated actions in order to complete a certain task. This is particularly true when two or more agents interact with each other in the same environment and need coordinate their behavior [2].

---

[*]The first two authors contributed equally to this work.

Focusing on rational behavior, correlations can manifest themselves even in unstructured domains, though at a higher level of abstraction of the decision-making process. This is because rationality itself implies the existence of an underlying objective optimized by the agent that represents the agent's intentions and incentivizes to choose one action over another. Typically, these goals persist at least for a short period of time, causing dependencies between consecutive action choices (Section 4.2).

In this paper, we propose a learning framework that offers a direct way to model such correlations in *finite* decision-making problems, i.e., involving systems with discrete state and action spaces. A key feature of our framework is that it allows to capture correlations at any level of the process, i.e., in the system environment, at the intentional level, or directly at the level of the executed actions. We encode the underlying structure in a hierarchical Bayesian model, for which we derive a tractable variational inference method based on Pólya-Gamma augmentation that allows a fully probabilistic treatment of the learning problem. Results on common benchmark problems and a queueing network simulation demonstrate the advantages of the framework. The accompanying code is publicly available via Git.[1]

### Related Work

Modeling correlations in decision-making is a common theme in reinforcement learning and related fields. Gaussian processes (GPs) offer a flexible tool for this purpose and are widely used in a broad variety of contexts. Moreover, movement primitives [18] provide an effective way to describe temporal relationships in control problems. However, the natural problem domain of both are continuous state-action environments, which lie outside the scope of this work.

Inferring correlation structure from count data has been discussed extensively in the context of topic modeling [13, 14] and factor analysis [29]. Recently, a GP classification algorithm with a scalable variational approach based on Pólya-Gamma augmentation was proposed [30]. Though these approaches are promising, they do not address the problem-specific modeling aspects of decision-making.

For agents acting in discrete environments, a number of customized solutions exist that allow to model specific characteristics of a decision-making problem. A broad class of methods that specifically target temporal correlations rely on hidden Markov models. Many of these approaches operate on the intentional level, modeling the temporal relationships of the different goals followed by the agent [22]. However, there also exist several approaches to capture spatial dependencies between these goals. For a recent overview, see [27] and the references therein. Dependencies on the action level have also been considered in the past but, like most intentional models, existing approaches largely focus on the temporal correlations in action sequences (such as probabilistic movement primitives [18]) or they are restricted to the special case of deterministic policies [26]. A probabilistic framework to capture correlations between discrete action distributions is described in [25].

When it comes to modeling transition dynamics, most existing approaches rely on GP models [4, 3]. In the Texplore method of [8], correlations within the transition dynamics are modeled with the help of a random forest, creating a mixture of decision tree outcomes. Yet, a full Bayesian description in form of an explicit prior distribution is missing in this approach. For behavior acquisition, prior distributions over transition dynamics are advantageous since they can easily be used in Bayesian reinforcement learning algorithms such as BEETLE [21] or BAMCP [6]. A particular example of a prior distribution over transition probabilities is given in [19] in the form of a Dirichlet mixture. However, the incorporation of prior knowledge expressing a particular correlation structure is difficult in this model.

To the best of our knowledge, there exists no principled method to explicitly model correlations in the transition dynamics of discrete environments. Also, a universally applicable inference tool for discrete environments, comparable to Gaussian processes, has not yet emerged. The goal of our work is to fill this gap by providing a flexible inference framework for such cases.

## 2   Background

### 2.1   Markov Decision Processes

In this paper, we consider finite Markov decision processes (MDPs) of the form $(\mathcal{S}, \mathcal{A}, \mathcal{T}, R)$, where $\mathcal{S} = \{1, \ldots, S\}$ is a finite state space containing $S$ distinct states, $\mathcal{A} = \{1, \ldots, A\}$ is an action space

comprising $A$ actions for each state, $\mathcal{T} : \mathcal{S} \times \mathcal{S} \times \mathcal{A} \rightarrow [0, 1]$ is the state transition model specifying the probability distribution over next states for each current state and action, and $R : \mathcal{S} \times \mathcal{A} \rightarrow \mathbb{R}$ is a reward function. For further details, see [28].

## 2.2 Inference in MDPs

In decision-making with discrete state and action spaces, we are often faced with integer-valued quantities modeled as draws from multinomial distributions, $\mathbf{x}_c \sim \mathrm{Mult}(\mathbf{x}_c \mid N_c, \mathbf{p}_c)$, $N_c \in \mathbb{N}$, $\mathbf{p}_c \in \Delta^K$, where $K$ denotes the number of categories, $c \in \mathcal{C}$ indexes some finite covariate space with cardinality $C$, and $\mathbf{p}_c$ parametrizes the distribution at a given covariate value $c$. Herein, $\mathbf{x}_c$ can either represent actual count data observed during an experiment or describe some latent variable of our model. For example, when modeling the policy of an agent in an MDP, $\mathbf{x}_c$ may represent the vector of action counts observed at a particular state, in which case $\mathcal{C} = \mathcal{S}$, $K = A$, and $N_c$ is the total number of times we observe the agent choosing an action at state $c$. Similarly, when modeling state transition probabilities, $\mathbf{x}_c$ could be the vector counting outgoing transitions from some state $s$ for a given action $a$ (in which case $\mathcal{C} = \mathcal{S} \times A$) or, when modeling the agent's intentions, $\mathbf{x}_c$ could describe the number of times the agent follows a particular goal, which itself might be unobservable (Section 4.2).

When facing the inverse problem of inferring the probability vectors $\{\mathbf{p}_c\}$ from the count data $\{\mathbf{x}_c\}$, a computationally straightforward approach is to model the probability vectors using independent Dirichlet distributions for all covariate values, i.e., $\mathbf{p}_c \sim \mathrm{Dir}(\mathbf{p}_c \mid \boldsymbol{\alpha}_c) \; \forall c \in \mathcal{C}$, where $\boldsymbol{\alpha}_c \in \mathbb{R}_{>0}^K$ is a local concentration parameter for covariate value $c$. However, the resulting model is agnostic to the rich correlation structure present in most MDPs (Section 1) and thus ignores much of the prior information we have about the underlying decision-making problem. A more natural approach would be to model the probability vectors $\{\mathbf{p}_c\}$ jointly using common prior model, in order to capture their dependency structure. Unfortunately, this renders exact posterior inference intractable, since the resulting prior distributions are no longer conjugate to the multinomial likelihood.

Recently, a method for approximate inference in dependent multinomial models has been developed to account for the inherent correlation of the probability vectors [14]. To this end, the following prior model was introduced,

$$\mathbf{p}_c = \mathbf{\Pi}_{\mathrm{SB}}(\boldsymbol{\psi}_{c\cdot}), \qquad \boldsymbol{\psi}_{\cdot k} \sim \mathcal{N}(\boldsymbol{\psi}_{\cdot k} \mid \boldsymbol{\mu}_k, \boldsymbol{\Sigma}), \; k = 1, \ldots, K - 1. \qquad (1)$$

Herein, $\mathbf{\Pi}_{\mathrm{SB}}(\boldsymbol{\zeta}) = [\Pi_{\mathrm{SB}}^{(1)}(\boldsymbol{\zeta}), \ldots, \Pi_{\mathrm{SB}}^{(K)}(\boldsymbol{\zeta})]^\top$ is the *logistic stick-breaking transformation*, where

$$\Pi_{\mathrm{SB}}^{(k)}(\boldsymbol{\zeta}) = \sigma(\zeta_k) \prod_{j<k}(1 - \sigma(\zeta_j)), \; k = 1, \ldots, K - 1, \qquad \Pi_{\mathrm{SB}}^{(K)}(\boldsymbol{\zeta}) = 1 - \sum_{k=1}^{K-1} \Pi_{\mathrm{SB}}^{(k)}(\boldsymbol{\zeta}),$$

and $\sigma$ is the logistic function. The purpose of this transformation is to map the real-valued Gaussian variables $\{\boldsymbol{\psi}_{\cdot k}\}$ to the simplex by passing each entry through the sigmoid function and subsequently applying a regular stick-breaking construction [10]. Through the correlation structure $\boldsymbol{\Sigma}$ of the latent variables $\boldsymbol{\Psi}$, the transformed probability vectors $\{\mathbf{p}_c\}$ become dependent. Posterior inference for the latent variables $\boldsymbol{\Psi}$ can be traced out efficiently by introducing a set of auxiliary Pólya-Gamma (PG) variables, which leads to a conjugate model in the augmented space. This enables a simple inference procedure based on blocked Gibbs sampling, where the Gaussian variables and the PG variables are sampled in turn, conditionally on each other and the count data $\{\mathbf{x}_c\}$.

In the following section, we present a variational inference (VI) [1] approach utilizing this augmentation trick, which establishes a closed-form approximation scheme for the posterior distribution. Moreover, we present a hyper-parameter optimization method based on variational expectation-maximization that allows us to calibrate our model to a particular problem type, avoiding the need for manual parameter tuning. Applied in combination, this takes us beyond existing sampling-based approaches, providing a fast and automated inference algorithm for correlated count data.

## 3 Variational Inference for Dependent Multinomial Models

The goal of our inference procedure is to approximate the posterior distribution $p(\boldsymbol{\Psi} \mid \mathbf{X})$, where $\mathbf{X} = [\mathbf{x}_1, \ldots, \mathbf{x}_C]$ represents the data matrix and $\boldsymbol{\Psi} = [\boldsymbol{\psi}_{\cdot 1}, \ldots, \boldsymbol{\psi}_{\cdot K-1}]$ is the matrix of real-valued latent parameters. Exact inference is intractable since the calculation of $p(\boldsymbol{\Psi} \mid \mathbf{X})$ requires

marginalization over the joint parameter space of all variables $\boldsymbol{\Psi}$. Instead of following a Monte Carlo approach as in [14], we resort to a variational approximation. To this end, we search for the best approximating distribution from a family of distributions $\mathcal{Q}_{\boldsymbol{\Psi}}$ such that

$$p(\boldsymbol{\Psi} \mid \mathbf{X}) \approx q^*(\boldsymbol{\Psi}) = \underset{q(\boldsymbol{\Psi}) \in \mathcal{Q}_{\boldsymbol{\Psi}}}{\arg \min} \mathrm{KL}\left(q(\boldsymbol{\Psi}) \parallel p(\boldsymbol{\Psi} \mid \mathbf{X})\right). \tag{2}$$

Carrying out this optimization under a multinomial likelihood is hard because it involves intractable expectations over the variational distribution. However, in the following we show that, analogously to the inference scheme of [14], a PG augmentation of $\boldsymbol{\Psi}$ makes the optimization tractable. To this end, we introduce a family of augmented posterior distributions $\mathcal{Q}_{\boldsymbol{\Psi},\boldsymbol{\Omega}}$ and instead consider the problem

$$p(\boldsymbol{\Psi}, \boldsymbol{\Omega} \mid \mathbf{X}) \approx q^*(\boldsymbol{\Psi}, \boldsymbol{\Omega}) = \underset{q(\boldsymbol{\Psi}, \boldsymbol{\Omega}) \in \mathcal{Q}_{\boldsymbol{\Psi},\boldsymbol{\Omega}}}{\arg \min} \mathrm{KL}\left(q(\boldsymbol{\Psi}, \boldsymbol{\Omega}) \parallel p(\boldsymbol{\Psi}, \boldsymbol{\Omega} \mid \mathbf{X})\right), \tag{3}$$

where $\boldsymbol{\Omega} = [\boldsymbol{\omega}_1, \ldots, \boldsymbol{\omega}_{K-1}] \in \mathbb{R}^{C \times K-1}$ denotes the matrix of auxiliary variables. Notice that the desired posterior can be recovered as the marginal $p(\boldsymbol{\Psi} \mid \mathbf{X}) = \int p(\boldsymbol{\Psi}, \boldsymbol{\Omega} \mid \mathbf{X}) \, d\boldsymbol{\Omega}$.

First, we note that solving the optimization problem is equivalent to maximizing the evidence lower bound (ELBO)

$$\log p(\mathbf{X}) \geq L(q) = \mathsf{E}\left[\log p(\boldsymbol{\Psi}, \boldsymbol{\Omega}, \mathbf{X})\right] - \mathsf{E}\left[\log q(\boldsymbol{\Psi}, \boldsymbol{\Omega})\right],$$

where the expectations are calculated w.r.t. the variational distribution $q(\boldsymbol{\Psi}, \boldsymbol{\Omega})$. In order to arrive at a tractable expression for the ELBO, we recapitulate the following data augmentation scheme derived in [14],

$$p(\boldsymbol{\Psi}, \mathbf{X}) = \left(\prod_{k=1}^{K-1} \mathcal{N}(\boldsymbol{\psi}_{\cdot k} \mid \boldsymbol{\mu}_k, \boldsymbol{\Sigma})\right) \left(\prod_{c=1}^{C} \mathrm{Mult}(\mathbf{x}_c \mid N_c, \boldsymbol{\Pi}_{\mathrm{SB}}(\boldsymbol{\psi}_{c \cdot}))\right)$$

$$= \left(\prod_{k=1}^{K-1} \mathcal{N}(\boldsymbol{\psi}_{\cdot k} \mid \boldsymbol{\mu}_k, \boldsymbol{\Sigma})\right) \left(\prod_{c=1}^{C} \prod_{k=1}^{K-1} \mathrm{Bin}(x_{ck} \mid b_{ck}, \sigma(\psi_{ck}))\right), \tag{4}$$

where the stick-breaking representation of the multinomial distribution has been expanded using $b_{ck} = N_c - \sum_{j<k} x_{cj}$. From Eq. (4), we arrive at

$$p(\boldsymbol{\Psi}, \mathbf{X}) = \left(\prod_{k=1}^{K-1} \mathcal{N}(\boldsymbol{\psi}_{\cdot k} \mid \boldsymbol{\mu}_k, \boldsymbol{\Sigma})\right) \left(\prod_{c=1}^{C} \prod_{k=1}^{K-1} \binom{b_{ck}}{x_{ck}} \sigma(\psi_{ck})^{x_{ck}} (1 - \sigma(\psi_{ck}))^{b_{ck}-x_{ck}}\right)$$

and the PG augmentation, as introduced in [20], is obtained using the integral identity

$$= \int \underbrace{\prod_{k=1}^{K-1} \mathcal{N}(\boldsymbol{\psi}_{\cdot k} \mid \boldsymbol{\mu}_k, \boldsymbol{\Sigma}) \prod_{c=1}^{C} \binom{b_{ck}}{x_{ck}} 2^{-b_{ck}} \exp(\kappa_{ck} \psi_{ck}) \exp(-\omega_{ck} \psi_{ck}^2/2) \, \mathrm{PG}(\omega_{ck} \mid b_{ck}, 0)}_{p(\boldsymbol{\Psi},\boldsymbol{\Omega},\mathbf{X})} \, d\boldsymbol{\Omega}.$$

Herein, $\kappa_{ck} = x_{ck} - b_{ck}/2$ and $\mathrm{PG}(\zeta \mid u, v)$ is the density of the Pólya-Gamma distribution, with the exponential tilting property $\mathrm{PG}(\zeta \mid u, v) = \frac{\exp(-\frac{v^2}{2}\zeta) \, \mathrm{PG}(\zeta|u,0)}{\cosh^{-u}(v/2)}$ and the first moment $\mathsf{E}\left[\zeta\right] = \frac{u}{2v} \tanh(v/2)$. With this augmented distribution at hand, we derive a mean-field approximation for the variational distribution as

$$q(\boldsymbol{\Psi}, \boldsymbol{\Omega}) = \prod_{k=1}^{K-1} q(\boldsymbol{\psi}_{\cdot k}) \prod_{c=1}^{C} q(\omega_{ck}).$$

Exploiting calculus of variations, we obtain the following parametric forms for the components of the variational distributions,

$$q(\boldsymbol{\psi}_{\cdot k}) = \mathcal{N}(\boldsymbol{\psi}_{\cdot k} \mid \boldsymbol{\lambda}_k, \boldsymbol{V}_k), \quad q(\omega_{ck}) = \mathrm{PG}(\omega_{ck} \mid b_{ck}, w_{ck}).$$

The optimal parameters and first moments of the variational distributions are

$$w_{ck} = \sqrt{\mathrm{E}[\psi_{ck}^2]}, \quad \mathbf{V}_k = (\boldsymbol{\Sigma}^{-1} + \mathrm{diag}\left(\mathrm{E}[\boldsymbol{\omega}_k]\right))^{-1}, \quad \boldsymbol{\lambda}_k = \mathbf{V}_k(\boldsymbol{\kappa}_k + \boldsymbol{\Sigma}^{-1}\boldsymbol{\mu}_k),$$

$$\mathrm{E}[\psi_{ck}^2] = \left((\mathbf{V}_k)_{cc} + \lambda_{ck}^2\right), \quad \mathrm{E}[\omega_{ck}] = \frac{b_{ck}}{2w_{ck}} \tanh(w_{ck}/2),$$

with $\boldsymbol{\omega}_k = [\omega_{1k}, \ldots, \omega_{Ck}]^\top$ and $\boldsymbol{\kappa}_k = [\kappa_{1k}, \ldots, \kappa_{Ck}]^\top$. A detailed derivation of the these results and the resulting ELBO is provided in Section A. The variational approximation can be optimized through coordinate-wise ascent by cycling through the parameters and their moments. The corresponding distribution over probability vectors $\{\mathbf{p}_c\}$ is defined implicitly through the deterministic relationship in Eq. (1).

**Hyper-Parameter Optimization**

For hyper-parameter learning, we employ a variational expectation-maximization approach [15] to optimize the ELBO after each update of the variational parameters. Assuming a covariance matrix $\boldsymbol{\Sigma}_{\boldsymbol{\theta}}$ parametrized by a vector $\boldsymbol{\theta} = [\theta_1, \ldots, \theta_J]^\top$, the ELBO can be written as

$$
L(q) = -\frac{K-1}{2}|\boldsymbol{\Sigma}_{\boldsymbol{\theta}}| + \frac{1}{2}\sum_{k=1}^{K-1}\log|\mathbf{V}_k| - \frac{1}{2}\sum_{k=1}^{K-1}\mathrm{tr}\,(\boldsymbol{\Sigma}_{\boldsymbol{\theta}}^{-1}\mathbf{V}_k)
$$
$$
-\frac{1}{2}\sum_{k=1}^{K-1}(\boldsymbol{\mu}_k - \boldsymbol{\lambda}_k)^\top \boldsymbol{\Sigma}_{\boldsymbol{\theta}}^{-1}(\boldsymbol{\mu}_k - \boldsymbol{\lambda}_k) + C(K-1) + \sum_{k=1}^{K-1}\sum_{c=1}^{C}\log\binom{b_{ck}}{x_{ck}}
$$
$$
-\sum_{k=1}^{K-1}\sum_{c=1}^{C}b_{ck}\log 2 + \sum_{k=1}^{K-1}\boldsymbol{\lambda}_k^\top\boldsymbol{\kappa}_k - \sum_{k=1}^{K-1}\sum_{c=1}^{C}b_{ck}\log\left(\cosh\frac{w_{ck}}{2}\right).
$$

A detailed derivation of this expression can be found in Section A.2. The corresponding gradients w.r.t. the hyper-parameters calculate to

$$
\frac{\partial L}{\partial \boldsymbol{\mu}_k} = \boldsymbol{\Sigma}_{\boldsymbol{\theta}}^{-1}(\boldsymbol{\mu}_k - \boldsymbol{\lambda}_k), \qquad \frac{\partial L}{\partial \theta_j} = -\frac{1}{2}\sum_{k=1}^{K-1}\left(\mathrm{tr}\,(\boldsymbol{\Sigma}_{\boldsymbol{\theta}}^{-1}\frac{\partial\boldsymbol{\Sigma}_{\boldsymbol{\theta}}}{\partial\theta_j}) - \mathrm{tr}\,(\boldsymbol{\Sigma}_{\boldsymbol{\theta}}^{-1}\frac{\partial\boldsymbol{\Sigma}_{\boldsymbol{\theta}}}{\partial\theta_j}\boldsymbol{\Sigma}_{\boldsymbol{\theta}}^{-1}\boldsymbol{V}_k)\right.
$$
$$
\left. -(\boldsymbol{\mu}_k - \boldsymbol{\lambda}_k)^\top\boldsymbol{\Sigma}_{\boldsymbol{\theta}}^{-1}\frac{\partial\boldsymbol{\Sigma}_{\boldsymbol{\theta}}}{\partial\theta_j}\boldsymbol{\Sigma}_{\boldsymbol{\theta}}^{-1}(\boldsymbol{\mu}_k - \boldsymbol{\lambda}_k)\right),
$$

which admits a closed-form solution for the optimal mean parameters, given by $\boldsymbol{\mu}_k = \boldsymbol{\lambda}_k$.

For the optimization of the covariance parameters $\boldsymbol{\theta}$, we can resort to a numerical scheme using the above gradient expression; however, this requires a full inversion of the covariance matrix in each update step. As it turns out, a closed-form expression can be found for the special case where $\boldsymbol{\theta}$ is a scale parameter, i.e., $\boldsymbol{\Sigma}_{\boldsymbol{\theta}} = \theta\tilde{\boldsymbol{\Sigma}}$, for some fixed $\tilde{\boldsymbol{\Sigma}}$. The optimal parameter value can then be determined as

$$
\theta = \frac{1}{KC}\sum_{k=1}^{K-1}\mathrm{tr}\,\left(\tilde{\boldsymbol{\Sigma}}^{-1}\left(\boldsymbol{V}_k + (\boldsymbol{\mu}_k - \boldsymbol{\lambda}_k)(\boldsymbol{\mu}_k - \boldsymbol{\lambda}_k)^\top\right)\right).
$$

The closed-form solution avoids repeated matrix inversions since $\tilde{\boldsymbol{\Sigma}}^{-1}$, being independent of all hyper-parameters and variational parameters, can be evaluated at the start of the optimization procedure. The full derivation of the gradients and the closed-form expression is provided in Section B.

For the experiments in the following section, we consider a squared exponential covariance function of the form $(\boldsymbol{\Sigma}_{\boldsymbol{\theta}})_{cc'} = \theta\exp\left(-\frac{d(c,c')^2}{l^2}\right)$, with a covariate distance measure $d : \mathcal{C} \times \mathcal{C} \to \mathbb{R}_{\geq 0}$ and a length scale $l \in \mathbb{R}_{\geq 0}$ adapted to the specific modeling scenario. Yet, we note that for model selection purposes multiple covariance functions can be easily compared against each other based on the resulting values of the ELBO [15]. Also, a combination of functions can be employed, provided that the resulting covariance matrix is positive semi-definite (see covariance kernels of GPs [23]).

## 4 Experiments

To demonstrate the versatility of our inference framework, we test it on a number of modeling scenarios that commonly occur in decision-making contexts. Due to space limitations, we restrict ourselves to imitation learning, subgoal modeling, system identification, and Bayesian reinforcement learning. However, we would like to point out that the same modeling principles can be applied in many other situations, e.g., for behavior coordination among agents [2] or knowledge transfer between related tasks [11], to name just two examples. A more comprehensive evaluation study is left for future work.

### 4.1 Imitation Learning

First, we illustrate our framework on an imitation learning example, where we aspire to reconstruct the policy of an agent (in this context called the *expert*) from observed behavior. For the reconstruction,

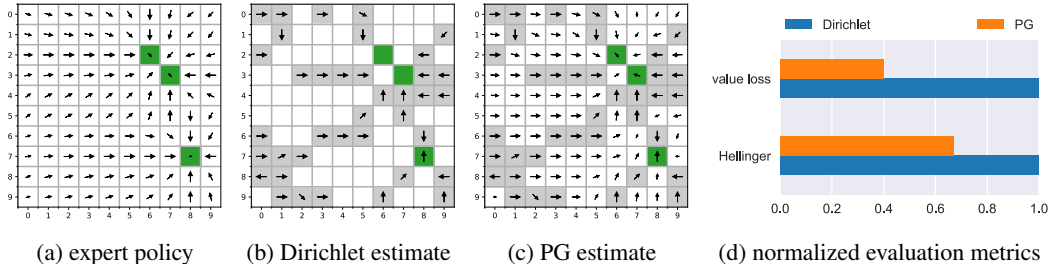

|                | (a) expert policy | (b) Dirichlet estimate | (c) PG estimate | (d) normalized evaluation metrics |

Figure 1: Imitation learning example. The expert policy in (a) is reconstructed using the posterior mean estimates of (b) an independent Dirichlet policy model and (c) a correlated PG model, based on action data observed at the states marked in gray. The PG joint estimate of the local policies yields a significantly improved reconstruction, as shown by the resulting Hellinger distance to the expert policy and the corresponding value loss [27] in (d).

we suppose to have access to a demonstration data set $\mathcal{D} = \{(s_d, a_d) \in \mathcal{S} \times \mathcal{A}\}_{d=1}^D$ containing $D$ state-action pairs, where each action has been generated through the expert policy, i.e., $a_d \sim \pi(a \mid s_d)$. Assuming a discrete state and action space, the policy can be represented as a stochastic matrix $\mathbf{\Pi} = [\boldsymbol{\pi}_1, \dots, \boldsymbol{\pi}_S]$, whose $i$th column $\boldsymbol{\pi}_i \in \Delta^A$ represents the local action distribution of the expert at state $i$ in form of a vector. Our goal is to estimate this matrix from the demonstrations $\mathcal{D}$. By constructing the count matrix $(\mathbf{X})_{ij} = \sum_{d=1}^D \mathbb{1}(s_d = i \wedge a_d = j)$, the inference problem can be directly mapped to our PG model, which allows to jointly estimate the coupled quantities $\{\boldsymbol{\pi}_i\}$ through their latent representation $\mathbf{\Psi}$ by approximating the posterior distribution $p(\mathbf{\Psi} \mid \mathbf{X})$ in Eq. (2). In this case, the covariate set $\mathcal{C}$ is described by the state space $\mathcal{S}$.

To demonstrate the advantages of this joint inference approach over a correlation-agnostic estimation method, we compare our framework to the independent Dirichlet model described in Section 2.2. Both reconstruction methods are evaluated on a classical grid world scenario comprising $S = 100$ states and $A = 4$ actions. Each action triggers a noisy transition in one of the four cardinal directions such that the pattern of the resulting next-state distribution resembles a discretized Gaussian distribution centered around the targeted adjacent state. Rewards are distributed randomly in the environment. The expert follows a near-optimal stochastic policy, choosing actions from a softmax distribution obtained from the Q-values of the current state. An example scenario is shown in Fig. 1a, where the the displayed arrows are obtained by weighting the four unit-length vectors associated with the action set $\mathcal{A}$ according to their local action probabilities. The reward locations are highlighted in green.

Fig. 1b shows the reconstruction of the policy obtained through the independent Dirichlet model. Since no dependencies between the local action distributions are considered in this approach, a posterior estimate can only be obtained for states where demonstration data is available, highlighted by the gray coloring of the background. For all remaining states, the mean estimate predicts a uniform action choice for the expert behavior since no action is preferred by the symmetry of the Dirichlet prior, resulting in an effective arrow length of zero. By contrast, the PG model (Fig. 1c) is able to generalize the expert behavior to unobserved regions of the state space, resulting in significantly improved reconstruction of the policy (Fig. 1d). To capture the underling correlations, we used the Euclidean distance between the grid positions as covariate distance measure $d$ and set $l$ to the maximum occurring distance value.

## 4.2 Subgoal Modeling

In many situations, modeling the actions of an agent is not of primary interest or proves to be difficult, e.g., because a more comprehensive understanding of the agent's behavior is desired (see inverse reinforcement learning [16] and preference elicitation [24]) or because the policy is of complex form due to intricate system dynamics. A typical example is robot object manipulation, where contact-rich dynamics can make it difficult for a controller trained from a small number of demonstrations to appropriately generalize the expert behavior [31]. A simplistic example illustrating this problem is depicted in Fig. 2a, where the agent behavior is heavily affected by the geometry of the environment and the action profiles at two wall-separated states differ drastically. Similarly to Section 4.1, we aspire to reconstruct the shown behavior from a demonstration data set of the form $\mathcal{D} = \{(s_d, a_d) \in \mathcal{S} \times \mathcal{A}\}_{d=1}^D$, depicted in Fig. 2b. This time, however, we follow a conceptually

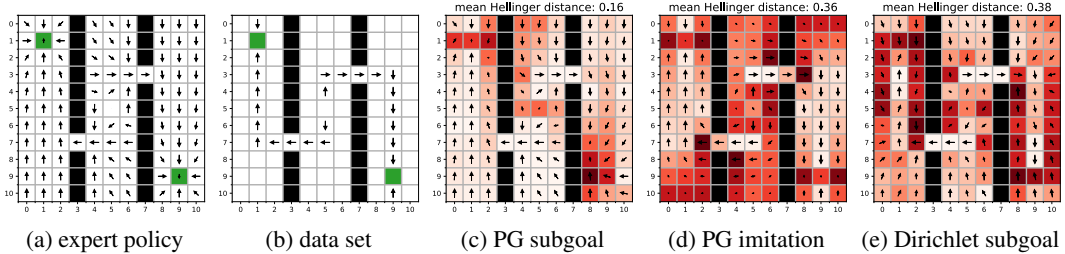

| | | | | |
|---|---|---|---|---|
| (a) expert policy | (b) data set | (c) PG subgoal | (d) PG imitation | (e) Dirichlet subgoal |

Figure 2: Subgoal modeling example. The expert policy in (a) targeting the green reward states is reconstructed from the demonstration data set in (b). By generalizing the demonstrations on the intentional level while taking into account the geometry of the problem, the PG subgoal model in (c) yields a significantly improved reconstruction compared to the corresponding action-based model in (d) and the uncorrelated subgoal model in (e). Red color encodes the Hellinger distance to the expert policy.

different line of reasoning and assume that each state $s \in \mathcal{S}$ has an associated subgoal $g_s$ that the agent is targeting at that state. Thus, action $a_d$ is considered as being drawn from some goal-dependent action distribution $p(a_d \mid s_d, g_{s_d})$. For our example, we adopt the normalized softmax action model described in [27]. Spatial relationships between the agent's decisions are taken into account with the help of our PG framework, by coupling the probability vectors that govern the underlying subgoal selection process, i.e., $g_s \sim \mathrm{Cat}(\mathbf{p}_s)$, where $\mathbf{p}_s$ is described through the stick-breaking construction in Eq. (1). Accordingly, the underlying covariate space of the PG model is $\mathcal{C} = \mathcal{S}$.

With the additional level of hierarchy introduced, the count data $\mathbf{X}$ to train our model is not directly available since the subgoals $\{g_s\}_{s=1}^S$ are not observable. For demonstration purposes, instead of deriving the full variational update for the extended model, we follow a simpler strategy that leverages the existing inference framework within a Gibbs sampling procedure, switching between variational updates and drawing posterior samples of the latent subgoal variables. More precisely, we iterate between 1) computing the variational approximation in Eq. (3) for a given set of subgoals $\{g_s\}_{s=1}^S$, treating each subgoal as single observation count, i.e., $\mathbf{x}_s = \mathrm{OneHot}(g_s) \sim \mathrm{Mult}(\mathbf{x}_s \mid N_s = 1, \mathbf{p}_s)$ and 2) updating the latent assignments based on the induced goal distributions, i.e., $g_s \sim \mathrm{Cat}(\Pi_{\mathrm{SB}}(\boldsymbol{\psi}_{s\cdot}))$.

Fig. 2c shows the policy model obtained by averaging the predictive action distributions of $M = 100$ drawn subgoal configurations, i.e., $\hat{\pi}(a \mid s) = \frac{1}{M} \sum_{m=1}^{M} p(a \mid s, g_s^{\langle m \rangle})$, where $g_s^{\langle m \rangle}$ denotes the $m$th Gibbs sample of the subgoal assignment at state $s$. The obtained reconstruction is visibly better than the one produced by the corresponding imitation learning model in Fig. 2d, which interpolates the behavior on the action level and thus fails to navigate the agent around the walls. While the Dirichlet-based subgoal model (Fig. 2e) can generally account for the walls through the use of the underlying softmax action model, it cannot propagate the goal information to unvisited states. For the considered uninformative prior distribution over subgoal locations, this has the consequence that actions assigned to such states have the tendency to transport the agent to the center of the environment, as this is the dominating move obtained when blindly averaging over all possible goal locations.

## 4.3 System Identification & Bayesian Reinforcement Learning

Having focused our attention on learning a model of an observed policy, we now enter the realm of Bayesian reinforcement learning (BRL) and optimize a behavioral model to the particular dynamics of a given environment. For this purpose, we slightly modify our grid world from Section 4.1 by placing a target reward of $+1$ in one corner and repositioning the agent to the opposite corner whenever the target state is reached (compare "Grid10" domain in [6]). For the experiment, we assume that the agent is aware of the target reward but does not know the transition dynamics of the environment.

### 4.3.1 System Identification

For the beginning, we ignore the reward mechanism altogether and focus on learning the transition dynamics of the environment. To this end, we let the agent perform a random walk on the grid, choosing actions uniformly at random and observing the resulting state transitions. The recorded state-action sequence $(s_1, a_1, s_2, a_2, \ldots, a_{T-1}, s_T)$ is summarized in the form of count matrices $[\mathbf{X}^{(1)}, \ldots, \mathbf{X}^{(A)}]$, where the element $(\mathbf{X}^{(a)})_{ij} = \sum_{t=1}^{T} \mathbb{1}(a_t = a \wedge s_t = i \wedge s_{t+1} = j)$ represents

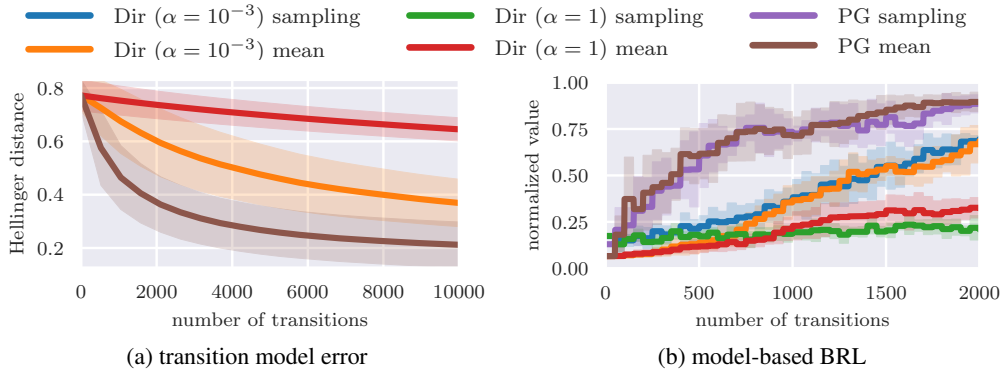

(a) transition model error        (b) model-based BRL

Figure 3: Bayesian reinforcement learning results. (a) Estimation error of the transition dynamics over the number of observed transitions. Shown are the Hellinger distances to the true next-state distribution and the standard deviation of the estimation error, both averaged over all states and actions of the MDP. (b) Expected returns of the learned policies (normalized by the optimal return) when replanning with the estimated transition dynamics after every fiftieth state transition.

the number of observed transitions from state $i$ to $j$ for action $a$. Analogously to the previous two experiments, we estimate the transition dynamics of the environment from these matrices using an independent Dirichlet prior model and our PG framework, where we employ a separate model for each transition count matrix. The resulting estimation accuracy is described by the graphs in Fig. 3a, which show the distance between the ground truth dynamics of the environment and those predicted by the models, averaged over all states and actions. As expected, our PG model significantly outperforms the naive Dirichlet approach.

### 4.3.2 Bayesian Reinforcement Learning

Next, we consider the problem of combined model-learning and decision-making by exploiting the experience gathered from previous system interactions to optimize future behavior. Bayesian reinforcement learning offers a natural playground for this task as it intrinsically balances the importance of information gathering and instantaneous reward maximization, avoiding the exploration-exploitation dilemma encountered in classical reinforcement learning schemes [5].

To determine the optimal trade-off between these two competing objectives computationally, we follow the principle of *posterior sampling for reinforcement learning* (PSRL) [17], where future actions are planned using a probabilistic model of the environment's transition dynamics. Herein, we consider two variants: (1) In the first variant, we compute the optimal Q-values for a fixed number of posterior samples representing instantiations of the transition model and choose the policy that yields the highest expected return on average. (2) In the second variant, we select the greedy policy dictated by the posterior mean of the transition dynamics. In both cases, the obtained policy is followed for a fixed number of transitions before new observations are taken into account for updating the posterior distribution. Fig. 3b shows the expected returns of the so-obtained policies over the entire execution period for the three prior models evaluated in Fig. 3a and both PSRL variants. The graphs reveal that the PG approach requires significantly fewer transitions to learn an effective decision-making strategy.

### 4.3.3 Queueing Network Modeling

As a final experiment, we evaluate our model on a network scheduling problem, depicted in Fig. 4a. The considered two-server network consists of two queues with buffer lengths $B_1 = B_2 = 10$. The state of the system is determined by the number of packets in each queue, summarized by the queueing vector $\mathbf{b} = [b_1, b_2]$, where $b_i$ denotes the number of packets in queue $i$. The underlying system state space is $\mathcal{S} = \{0, \dots, B_1\} \times \{0, \dots, B_2\}$ with size $S = (B_1 + 1)(B_2 + 1)$.

For our experiment, we consider a system with batch arrivals and batch servicing. The task for the agent is to schedule the traffic flow of the network under the condition that only one of the queues can be processed at a time. Accordingly, the actions are encoded as $a = 1$ for serving queue 1 and $a = 2$ for serving queue 2. The number of packets arriving at queue 1 is modeled as

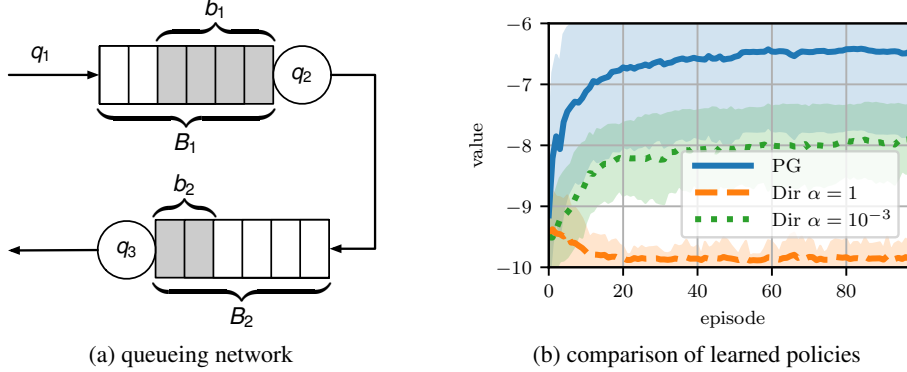

(a) queueing network            (b) comparison of learned policies

Figure 4: BRL for batch queueing. (a) Considered two-server queueing network. (b) Expected returns over the number of learning episodes, each consisting of twenty state transitions.

$q_1 \sim \mathrm{Pois}(q_1 \mid \vartheta_1)$ with mean rate $\vartheta_1 = 1$. The packets are transferred to buffer 1 and subsequently processed in batches of random size $q_2 \sim \mathrm{Pois}(q_2 \mid \vartheta_2)$, provided that the agent selects queue 1. Therefore, $\vartheta_2 = \beta_1 \mathbb{1}(a = 1)$, where we consider an average batch size of $\beta_1 = 3$. Processed packets are transferred to the second queue, where they wait to be processed further in batches of size $q_3 \sim \mathrm{Pois}(q_3 \mid \vartheta_3)$, with $\vartheta_3 = \beta_2 \mathbb{1}(a = 2)$ and an average batch size of $\beta_2 = 2$. The resulting transition to the new queueing state $\mathbf{b}'$ after one processing step can be compactly written as $\mathbf{b}' = [(b_1 + q_1 - q_2)_0^{B_1}, (b_2 + q_2 - q_3)_0^{B_2}]$, where the truncation operation $(\cdot)_0^B = \max(0, \min(B, \cdot))$ accounts for the nonnegativity and finiteness of the buffers. The reward function, which is known to the agent, computes the negative sum of the queue lengths $R(\mathbf{b}) = -(b_1 + b_2)$. Despite the simplistic architecture of the network, finding an optimal policy for this problem is challenging since determining the state transition matrices requires nontrivial calculations involving concatenations of Poisson distributions. More importantly, when applied in a real-world context, the arrival and processing rates of the network are typically unknown so that planning-based methods cannot be applied.

Fig. 4b shows the evaluation of PSRL on the network. As in the previous experiment, we use a separate PG model for each action and compute the covariance matrix $\boldsymbol{\Sigma}_{\boldsymbol{\theta}}$ based on the normalized Euclidean distances between the queueing states of the system. This encodes our prior knowledge that the queue lengths obtained after servicing two independent copies of the network tend to be similar if their previous buffer states were similar. Our agent follows a greedy strategy w.r.t. the posterior mean of the estimated model. The policy is evaluated after each policy update by performing one thousand steps from all possible queueing states of the system. As the graphs reveal, the PG approach significantly outperforms its correlation agnostic counterpart, requiring fewer interactions with the system while yielding better scheduling strategies by generalizing the networks dynamics over queueing states.

## 5 Conclusion

With the proposed variational PG model, we have presented a self-contained learning framework for flexible use in many common decision-making contexts. The framework allows an intuitive consideration of prior knowledge about the behavior of an agent and the structures of its environment, which can significantly boost the predictive performance of the resulting models by leveraging correlations and reoccurring patterns in the decision-making process. A key feature is the adjustment of the model regularization through automatic calibration of its hyper-parameters to the specific decision-making scenario at hand, which provides a built-in solution to infer the effective range of correlations from the data. We have evaluated the framework on various benchmark tasks including a realistic queueing problem, which in a real-world situation admits no planning-based solution due to unknown system parameters. In all presented scenarios, our framework consistently outperformed the naive baseline methods, which neglect the rich statistical relationships to be unraveled in the estimation problems.

**Acknowledgments**

This work has been funded by the German Research Foundation (DFG) as part of the projects B4 and C3 within the Collaborative Research Center (CRC) 1053 – MAKI.

## Footnotes

[1] https://git.rwth-aachen.de/bcs/correlation_priors_for_rl

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
