[Supplementary Material]

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

## A   Optimizing the Evidence Lower Bound (ELBO)

In this section, we provide the detailed calculation for optimizing the ELBO of the presented model. First, we derive the optimal variational distributions and their parameters in Section A.1. Then, we derive the ELBO as function of the variational parameters in Section A.2.

The ELBO for the proposed model with PG augmentation scheme is

$$L(q) = \mathsf{E}\left[\log p(\boldsymbol{\Psi}, \boldsymbol{\Omega}, \mathbf{X})\right] - \mathsf{E}\left[\log q(\boldsymbol{\Psi}, \boldsymbol{\Omega})\right]. \tag{5}$$

The joint distribution of the model is given by

$$
\begin{aligned}
p(\boldsymbol{\Psi}, \boldsymbol{\Omega}, \mathbf{X}) = &\prod_{k=1}^{K-1} \mathcal{N}(\boldsymbol{\psi}_{\cdot k} \mid \boldsymbol{\mu}_k, \boldsymbol{\Sigma}) \\
&\times \prod_{c=1}^{C} \binom{b_{ck}}{x_{ck}} 2^{-b_{ck}} \exp(\kappa_{ck}\psi_{ck}) \exp(-\omega_{ck}\psi_{ck}^2/2)\, \mathrm{PG}(\omega_{ck} \mid b_{ck}, 0).
\end{aligned}
\tag{6}
$$

For the derivation, we assume a factorized approximation with

$$q(\boldsymbol{\Psi}, \boldsymbol{\Omega}) = \prod_{k=1}^{K-1} q(\boldsymbol{\psi}_{\cdot k}) \prod_{c=1}^{C} q(\omega_{ck}). \tag{7}$$

### A.1   Parameteric Forms of the Variational Distributions

**Calculating the Variational Distribution** $q(\boldsymbol{\psi}_{\cdot k})$

First, we calculate the optimal forms of the variational distributions $q(\boldsymbol{\psi}_{\cdot k})$, $k = 1, \ldots, K-1$. Collecting all terms in Eq. (5) that depend on $\boldsymbol{\psi}_{\cdot k}$ gives

$$L(q) = L(q(\boldsymbol{\psi}_{\cdot k})) + L_{\mathrm{const}}.$$

Due to the factorization in Eq. (7) together with Eq. (6), we have

$$L(q(\boldsymbol{\psi}_{\cdot k})) = \mathsf{E}\left[\log \mathcal{N}(\boldsymbol{\psi}_{\cdot k} \mid \boldsymbol{\mu}_k, \boldsymbol{\Sigma})\right] + \sum_{c=1}^{C} \mathsf{E}\left[\psi_{ck}\right]\kappa_{ck} - \sum_{c=1}^{C} \mathsf{E}\left[\omega_{ck}\psi_{ck}^2/2\right] - \mathsf{E}\left[\log q(\boldsymbol{\psi}_{\cdot k})\right].$$

The optimal distribution can be calculated by introducing the Lagrangian

$$
\begin{aligned}
\mathcal{L}(q(\boldsymbol{\psi}_{\cdot k}), \nu) = &\mathsf{E}\left[\log \mathcal{N}(\boldsymbol{\psi}_{\cdot k} \mid \boldsymbol{\mu}_k, \boldsymbol{\Sigma})\right] + \sum_{c=1}^{C} \mathsf{E}\left[\psi_{ck}\right]\kappa_{ck} - \sum_{c=1}^{C} \mathsf{E}\left[\omega_{ck}\psi_{ck}^2/2\right] \\
&- \mathsf{E}\left[\log q(\boldsymbol{\psi}_{\cdot k})\right] + \nu\left(\int q(\boldsymbol{\psi}_{\cdot k})\, d\boldsymbol{\psi}_{\cdot k} - 1\right),
\end{aligned}
$$

which ensures that $q(\boldsymbol{\psi}_{\cdot k})$ is a proper density, using $\nu$ as a Lagrange multiplier to enforce the normalization constraint. The Euler Lagrange equation and optimality condition are

$$\frac{\delta \mathcal{L}}{\delta q} = 0, \quad \frac{\partial \mathcal{L}}{\partial \nu} = 0. \tag{8}$$

The functional derivative of the Lagrangian yields

$$\frac{\delta \mathcal{L}}{\delta q} = \log \mathcal{N}(\boldsymbol{\psi}_{\cdot k} \mid \boldsymbol{\mu}_k, \boldsymbol{\Sigma}) + \sum_{c=1}^{C} \psi_{ck} \kappa_{ck} - \sum_{c=1}^{C} \mathsf{E}\left[\omega_{ck}\right] \psi_{ck}^2 / 2 - \log q(\boldsymbol{\psi}_{\cdot k}) - 1 + \nu.$$

By solving the Euler Lagrange equation for $q(\boldsymbol{\psi}_{\cdot k})$, we obtain

$$q(\boldsymbol{\psi}_{\cdot k}) = \exp\left(\nu - 1 + \log \mathcal{N}(\boldsymbol{\psi}_{\cdot k} \mid \boldsymbol{\mu}_k, \boldsymbol{\Sigma}) + \sum_{c=1}^{C} \psi_{ck} \kappa_{ck} - \sum_{c=1}^{C} \mathsf{E}\left[\omega_{ck}\right] \psi_{ck}^2 / 2\right).$$

The optimality condition (normalization constraint) in Eq. (8) yields

$$q(\boldsymbol{\psi}_{\cdot k}) \propto \mathcal{N}(\boldsymbol{\psi}_{\cdot k} \mid \boldsymbol{\mu}_k, \boldsymbol{\Sigma}) \exp\left(-\frac{1}{2} \sum_{c=1}^{C} \mathsf{E}\left[\omega_{ck}\right] \psi_{ck}^2 + \sum_{c=1}^{C} \psi_{ck} \kappa_{ck}\right)$$

$$\propto \mathcal{N}(\boldsymbol{\psi}_{\cdot k} \mid \boldsymbol{\mu}_k, \boldsymbol{\Sigma}) \prod_{c=1}^{C} \mathcal{N}(\frac{\kappa_{ck}}{\mathsf{E}\left[\omega_{ck}\right]} \mid \psi_{ck}, 1/\mathsf{E}\left[\omega_{ck}\right])$$

$$= \mathcal{N}(\boldsymbol{\psi}_{\cdot k} \mid \boldsymbol{\mu}_k, \boldsymbol{\Sigma}) \mathcal{N}(\text{diag}\left(\mathsf{E}\left[\boldsymbol{\omega}_k\right]\right)^{-1} \boldsymbol{\kappa}_k \mid \boldsymbol{\psi}_{\cdot k}, \text{diag}\left(\mathsf{E}\left[\boldsymbol{\omega}_k\right]\right)^{-1}).$$

Therefore, the optimal distribution $q(\boldsymbol{\psi}_{\cdot k})$ can be identified as a Gaussian by completing the square

$$q(\boldsymbol{\psi}_{\cdot k}) = \mathcal{N}(\boldsymbol{\psi}_{\cdot k} \mid \boldsymbol{\lambda}_k, \mathbf{V}_k), \tag{9}$$

with the variational parameters

$$\mathbf{V}_k = (\boldsymbol{\Sigma}^{-1} + \text{diag}\left(\mathsf{E}\left[\boldsymbol{\omega}_k\right]\right))^{-1}, \quad \boldsymbol{\lambda}_k = \mathbf{V}_k(\boldsymbol{\kappa}_k + \boldsymbol{\Sigma}^{-1} \boldsymbol{\mu}_k). \tag{10}$$

**Calculating the Variational Distribution** $q(\omega_{ck})$

The distribution for $q(\omega_{ck})$ is calculated in a similar fashion. The ELBO in Eq. (5) in terms dependent on $\omega_{ck}$ can be written as

$$L(q(\omega_{ck})) = L(q(\omega_{ck})) + L_{\text{const}},$$

with

$$L(q(\omega_{ck})) = -\mathsf{E}\left[\omega_{ck}\right] \mathsf{E}\left[\psi_{ck}^2\right]/2 + \mathsf{E}\left[\log \text{PG}(\omega_{ck} \mid b_{ck}, 0)\right] - \mathsf{E}\left[\log q(\omega_{ck})\right].$$

The Lagrangian for the distribution $q(\omega_{ck})$ is

$$\mathcal{L}(q(\omega_{ck}), \nu) = -\mathsf{E}\left[\omega_{ck}\right] \mathsf{E}\left[\psi_{ck}^2\right]/2 + \mathsf{E}\left[\log \text{PG}(\omega_{ck} \mid b_{ck}, 0)\right] - \mathsf{E}\left[\log q(\omega_{ck})\right]$$
$$+ \nu \left(\int q(\omega_{ck}) \, d\omega_{ck} - 1\right)$$

The functional derivative of the Lagrangian yields

$$\frac{\delta \mathcal{L}}{\delta q} = -\omega_{ck} \mathsf{E}\left[\psi_{ck}^2\right]/2 + \log \text{PG}(\omega_{ck} \mid b_{ck}, 0) - \log q(\omega_{ck}) - 1 + \nu.$$

Solving the Euler Lagrange equation $\frac{\delta \mathcal{L}}{\delta q} = 0$ for $q(\omega_{ck})$, we find

$$q(\omega_{ck}) = \exp\left(\nu - 1 + \log \text{PG}(\omega_{ck} \mid b_{ck}, 0) - \omega_{ck} \mathsf{E}\left[\psi_{ck}^2\right]/2\right).$$

The normalization constraint is used to identify

$$q(\omega_{ck}) \propto \text{PG}(\omega_{ck} \mid b_{ck}, 0) \exp\left(-\omega_{ck} \mathsf{E}\left[\psi_{ck}^2\right]/2\right).$$

By exploiting the exponential tilting property of the PG distribution, that is,

$$\text{PG}(\zeta \mid u, v) \propto \exp(-\frac{v^2}{2}\zeta) \, \text{PG}(\zeta \mid u, 0),$$

we obtain

$$q(\omega_{ck}) = \text{PG}(\omega_{ck} \mid b_{ck}, w_{ck}), \tag{11}$$

with the variational parameter $w_{ck} = \sqrt{\mathsf{E}\left[\psi_{ck}^2\right]}$.

## A.2 The ELBO for the Optimal Distributions

The ELBO

$$L(q) = \mathsf{E}\left[\log p(\boldsymbol{\Psi}, \boldsymbol{\Omega}, \mathbf{X})\right] - \mathsf{E}\left[\log q(\boldsymbol{\Psi}, \boldsymbol{\Omega})\right]$$

in terms of the previously derived optimal distributions calculates to

$$\mathsf{E}\left[\log p(\boldsymbol{\Psi}, \boldsymbol{\Omega}, \mathbf{X})\right] = \sum_{k=1}^{K-1} \mathsf{E}\left[\log \mathcal{N}(\boldsymbol{\psi}_{\cdot k} \mid \boldsymbol{\mu}_k, \boldsymbol{\Sigma})\right] + \sum_{k=1}^{K-1} \sum_{c=1}^{C} \log \binom{b_{ck}}{x_{ck}}$$

$$- \sum_{k=1}^{K-1} \sum_{c=1}^{C} b_{ck} \log 2 + \sum_{k=1}^{K-1} \boldsymbol{\lambda}_k^{\top} \boldsymbol{\kappa}_k$$

$$+ \sum_{k=1}^{K-1} \sum_{c=1}^{C} \mathsf{E}\left[\log \mathrm{PG}(\omega_{ck} \mid b_{ck}, w_{ck})\right]$$

$$- \sum_{k=1}^{K-1} \sum_{c=1}^{C} b_{ck} \log \left(\cosh \frac{w_{ck}}{2}\right),$$

$$\mathsf{E}\left[\log q(\boldsymbol{\Psi}, \boldsymbol{\Omega})\right] = \sum_{k=1}^{K-1} \mathsf{E}\left[\log \mathcal{N}(\boldsymbol{\psi}_{\cdot k} \mid \boldsymbol{\lambda}_k, \mathbf{V}_k)\right] + \sum_{k=1}^{K-1} \sum_{c=1}^{C} \mathsf{E}\left[\log \mathrm{PG}(\omega_{ck} \mid b_{ck}, w_{ck})\right].$$

Canceling out the terms $\mathsf{E}\left[\log \mathrm{PG}(\omega_{ck} \mid b_{ck}, w_{ck})\right]$ and rewriting the prior and variational terms as Kullback-Leibler (KL) divergence, we obtain

$$L(q) = - \sum_{k=1}^{K-1} \mathrm{KL}\left(\mathcal{N}(\boldsymbol{\psi}_{\cdot k} \mid \boldsymbol{\lambda}_k, \mathbf{V}_k) \parallel \mathcal{N}(\boldsymbol{\psi}_{\cdot k} \mid \boldsymbol{\mu}_k, \boldsymbol{\Sigma})\right) + \sum_{k=1}^{K-1} \sum_{c=1}^{C} \log \binom{b_{ck}}{x_{ck}}$$

$$- \sum_{k=1}^{K-1} \sum_{c=1}^{C} b_{ck} \log 2 + \sum_{k=1}^{K-1} \boldsymbol{\lambda}_k^{\top} \boldsymbol{\kappa}_k - \sum_{k=1}^{K-1} \sum_{c=1}^{C} b_{ck} \log \left(\cosh \frac{w_{ck}}{2}\right).$$

Finally, by computing the KL divergence, the ELBO can be expressed in terms of the variational parameters as

$$L(q) = - \frac{K-1}{2} |\boldsymbol{\Sigma}| + \frac{1}{2} \sum_{k=1}^{K-1} \log |\mathbf{V}_k| - \frac{1}{2} \sum_{k=1}^{K-1} \mathrm{tr}\left(\boldsymbol{\Sigma}^{-1} \mathbf{V}_k\right)$$

$$- \frac{1}{2} \sum_{k=1}^{K-1} (\boldsymbol{\mu}_k - \boldsymbol{\lambda}_k)^{\top} \boldsymbol{\Sigma}^{-1} (\boldsymbol{\mu}_k - \boldsymbol{\lambda}_k) + C(K-1) + \sum_{k=1}^{K-1} \sum_{c=1}^{C} \log \binom{b_{ck}}{x_{ck}} \qquad (12)$$

$$- \sum_{k=1}^{K-1} \sum_{c=1}^{C} b_{ck} \log 2 + \sum_{k=1}^{K-1} \boldsymbol{\lambda}_k^{\top} \boldsymbol{\kappa}_k - \sum_{k=1}^{K-1} \sum_{c=1}^{C} b_{ck} \log \left(\cosh \frac{w_{ck}}{2}\right).$$

## B  Hyper-Parameter Optimization

In this section, we provide a derivation for the optimization of the hyper-parameters. By maximizing the ELBO $L(q)$ w.r.t. the parameters $\boldsymbol{\mu}_k$ and the parameterized covariance matrix $\boldsymbol{\Sigma}_{\boldsymbol{\theta}}$, we obtain equations for the variational expectation maximization algorithm.

The ELBO as a function of the mean $\boldsymbol{\mu}_k$ and covariance $\boldsymbol{\Sigma}_{\boldsymbol{\theta}}$ can be written as

$$L(\boldsymbol{\Sigma}_{\boldsymbol{\theta}}, \boldsymbol{\mu}_k) = - \frac{K-1}{2} |\boldsymbol{\Sigma}_{\boldsymbol{\theta}}| - \frac{1}{2} \sum_{k=1}^{K-1} \mathrm{tr}\left(\boldsymbol{\Sigma}_{\boldsymbol{\theta}}^{-1} \mathbf{V}_k\right)$$

$$- \frac{1}{2} \sum_{k=1}^{K-1} (\boldsymbol{\mu}_k - \boldsymbol{\lambda}_k)^{\top} \boldsymbol{\Sigma}_{\boldsymbol{\theta}}^{-1} (\boldsymbol{\mu}_k - \boldsymbol{\lambda}_k) + L_{\mathrm{const}}.$$

$$(13)$$

## B.1 Derivation of the Optimal Value for the Mean $\mu_k$

We calculate the gradient of Eq. (13) as

$$\frac{\partial L}{\partial \boldsymbol{\mu}_k} = \boldsymbol{\Sigma}_{\boldsymbol{\theta}}^{-1} \left( \boldsymbol{\mu}_k - \boldsymbol{\lambda}_k \right).$$

Setting the gradient to zero, we obtain

$$\boldsymbol{\mu}_k = \boldsymbol{\lambda}_k. \tag{14}$$

## B.2 Derivation of the Optimal Hyper-Parameters of $\boldsymbol{\Sigma}_{\boldsymbol{\theta}}$

We calculate the gradient of Eq. (13) as

$$
\begin{aligned}
\frac{\partial L}{\partial \theta_j} = &-\frac{K-1}{2} \operatorname{tr}\left(\boldsymbol{\Sigma}_{\boldsymbol{\theta}}^{-1} \frac{\partial \boldsymbol{\Sigma}_{\boldsymbol{\theta}}}{\partial \theta_j}\right) + \frac{1}{2} \sum_{k=1}^{K-1} \operatorname{tr}\left(\boldsymbol{\Sigma}_{\boldsymbol{\theta}}^{-1} \frac{\partial \boldsymbol{\Sigma}_{\boldsymbol{\theta}}}{\partial \theta_j} \boldsymbol{\Sigma}_{\boldsymbol{\theta}}^{-1} \boldsymbol{V}_k\right) \\
&+ \frac{1}{2} \sum_{k=1}^{K-1} (\boldsymbol{\mu}_k - \boldsymbol{\lambda}_k)^\top \boldsymbol{\Sigma}_{\boldsymbol{\theta}}^{-1} \frac{\partial \boldsymbol{\Sigma}_{\boldsymbol{\theta}}}{\partial \theta_j} \boldsymbol{\Sigma}_{\boldsymbol{\theta}}^{-1} (\boldsymbol{\mu}_k - \boldsymbol{\lambda}_k).
\end{aligned}
\tag{15}
$$

When considering the special case of a scaled covariance matrix $\boldsymbol{\Sigma}_{\boldsymbol{\theta}} = \theta \tilde{\boldsymbol{\Sigma}}$, we can find the optimizing hyper-parameter $\theta$ in closed form. Note that $\frac{\partial \boldsymbol{\Sigma}_{\boldsymbol{\theta}}}{\partial \theta} = \tilde{\boldsymbol{\Sigma}}$ and $\boldsymbol{\Sigma}_{\boldsymbol{\theta}}^{-1} = \frac{1}{\theta} \tilde{\boldsymbol{\Sigma}}^{-1}$. Therefore, the gradient computes to

$$\frac{\partial L}{\partial \theta} = -\frac{K}{2\theta} \operatorname{tr}\left(\mathbf{I}\right) + \frac{1}{2\theta^2} \sum_{k=1}^{K-1} \operatorname{tr}\left(\tilde{\boldsymbol{\Sigma}}^{-1} \boldsymbol{V}_k\right) + \frac{1}{2\theta^2} \sum_{k=1}^{K-1} (\boldsymbol{\mu}_k - \boldsymbol{\lambda}_k)^\top \tilde{\boldsymbol{\Sigma}}^{-1} (\boldsymbol{\mu}_k - \boldsymbol{\lambda}_k).$$

Setting the derivative to zero and solving for $\theta$, we obtain the closed-form expression

$$\theta = \frac{1}{KS} \sum_{k=1}^{K-1} \operatorname{tr}\left(\tilde{\boldsymbol{\Sigma}}^{-1} \left(\boldsymbol{V}_k + (\boldsymbol{\mu}_k - \boldsymbol{\lambda}_k)(\boldsymbol{\mu}_k - \boldsymbol{\lambda}_k)^\top\right)\right). \tag{16}$$