[Reviews · NeurIPS 2019]

Reviewer 1



The paper addresses the issue of exploiting correlation structures in Markov Decision Processes with discrete state spaces. The authors identify a gap that currently makes working with discrete state spaces problematic - that there is no principled method for modelling the state correlations that is flexible enough to accommodate all the ways in which these correlations could be exploited. The paper presents a hierarchical Bayesian model and proposes a variational inference method to find solutions. The model and procedure presented in the paper are an original application of variational inference, and represent a more general method for dealing with correlation structures than anything I have encountered before. The authors have done a great job of demonstrating this by employing three vastly different problem domains. It is unusual to see Imitation Learning, System Identification and Reinforcement Learning all being tested under a new model in one paper. All of these domains gain an advantage from modelling the correlations and from the inference procedure presented. The paper is very well written and easy to follow. It is unavoidable that some of the derivations are hard to read, and there were some portions that I had to take on faith but this is more a weakness in my prior knowledge than any weakness in the presentation. There is a significant weak point in the paper, and that is the baselines used in the experiments. I think the choice of DP as the general baseline makes sense to the authors as they are demonstrating that explicitly modelling the correlations yields an improvement in the observed performance. However, I think the conclusion is then much weaker as it is only able to show that modelling correlations explicitly is better than not modelling correlations at all. However, many existing methods implicity model these correlations. For instance, in the imitation learning setting, the DP model can only imitate the demonstrator in states that appear in the demonstrations. However, this is not generally a limitation in imitation learning, with methods like IRL being capable of learning behaviour in unobserved states. Similarly, in the System Identification portion, the DP model may be unrealistically naive. I think it would be interesting to see a comparison to a factored classifier, like the model learning portion of Texplore (RL-DT), which uses decision forests to generalise from observed dynamics. It could be argued that these comparisons are unfair as these methods are specific to their domain rather than being a general method for modelling correlations, but to people considering using the model, it is important to know whether the approach is superior to practical alternatives. Overall, I think this is paper was one that I was glad to spend time on and as such I think it would be of interest to the RL community in general. ----- Post rebuttal comments: thank you for your response. I'd really urge you to try make these changes, as I think it will greatly strengthen the paper, but I still think it is an interesting read.

Reviewer 2



The paper promises a principled solution to account for correlation in discrete state-action domains based on Poly-Gamma augmentation. The framework is empirically evaluated in three different settings: imitation learning, system identification & Bayesian RL. In each of these settings, the proposed method achieves a significant performance improvement over a correlation-agnostic baseline. Adding useful priors to RL tasks is highly non-trivial and can boost performance significantly. Hence, I was very interested in reading this paper. While the finite-state assumption is fairly restrictive, the generality of the framework, demonstrated by the different settings the method is tested on makes this potentially an interesting method to consider. That being said, what reduces the value of the paper considerably in my view is that it is insufficiently clear how useful the (family of) bias(es) is that can be encoded by this approach. The experiments mainly revolve around a very simple grid-based task. What if the example contains walls and the task becomes more of a navigation task: would the prior still be useful? It is possible that I have misunderstood something, so I will carefully read the author rebuttal and update my score accordingly. ==================================== The author rebuttal was helpful, although it did not fully resolve my concerns. Specifically, this sentence "the proposed methodology can be beneficial in all scenarios that contain some form of structure" is a little troublesome to me. What does "can" mean in this context? It doesn't clarify in which situations it will be beneficial and in which it won't. If the suggestion is that it will be beneficial in all situations that have structure, I'm not sure that I belief that. I still consider this a borderline paper, but will upgrade my score to a 6.

Reviewer 3



The paper proposes a method to model environment correlations during learning in discrete Markov decision processes. The framework applies variational inference in a hierarchical Bayesian framework. The proposed method is demonstrated on some small experiments in imitation learning, model learning and bayesian reinforcement learning settings. -The derivations in the paper seem to mostly be an application of the methods developed in reference [12] and standard methods from variational inference. The main novelty seems to be the application to MDP settings. -The paper gives a clear step-by-step derivation of method. -The paper provides a series of simple, but clear experiments to demonstrate the benefits of the method. -2 of the example experiments are small toy grid worlds, the last experiment is a slightly more involved queueing example.These experiments show benefits of the method in different settings and demonstrate that the proposed method outperforms a naive baseline. While the queuing example might be more difficult, all experiments seem limited to small scale toy domains. -For the related work, it would be nice to also contrast the approach in this paper with the work by Poupart on including prior knowledge in discrete Bayesian RL using mixtures of Dirichlets. *Poupart, Pascal, et al. An analytic solution to discrete Bayesian reinforcement learning. ICML, 2006. *Pavlov & Poupart. Towards global reinforcement learning. 2008

Reviewer 4



[Update after feedback round: since the authors did not have the chance to respond to my issues, my review should perhaps be down-weighted a bit. I have read the other reviews and the author feedback and was pleased to see that nonetheless some of my issues were (partially) addressed, in particular another experiment with slightly more complex dynamics (maze with walls). I think that the VI scheme derived in the paper is promising and that there are fairly straightforward ways to speed up / approximate the matrix inversion to improve scalability of the method at least to some degree. Nonetheless I personally think that the paper would benefit from a another round of revision, after which it could potentially be a much stronger paper. To me, the main hurdle for this is a comparison against the Gibbs-sampling version of the method. The paper should address the following questions: (I) why derive a VI scheme in the first place, where does the Gibbs-sampling method fail (scalability? performance? data-efficiency?), (II) the VI scheme makes additional approximations (e.g. mean-field) - how do they affect the solution class (theoretically and empirically), how does training of the model behave (local optima, mode-seeking/-covering behavior, convergence behavior), (III) an empirical comparison of the same questions as (II) but compared against the Gibbs-sampling method. Paired with a larger-scale experiment and perhaps a comparison against another method (as a bonus), I think that the paper would be very strong (particularly, as the authors point out, since the method is quite broadly applicable).] The paper addresses the problem of inferring “structure” via hierarchical probabilistic models in discrete domains. Structure refers to higher-order statistical regularities which can be expressed as priors and hyper-priors in hierarchical Bayesian models. In the discrete domain this typically leads to intractable inference or overly simplistic models that cannot capture complex higher-order regularities because of overly restrictive independence assumptions. The paper builds on previous work to alleviate this problem via the introduction of Polya-Gamma auxiliary variables. While the previous method allowed for efficient approximate inference via blocked Gibbs sampling, the main novelty of this paper is to introduce a suitable parametric model for the approximate posterior and derive a corresponding variational inference scheme that allows for closed-form expressions. Additionally, the paper derives (closed-form) update equations for an EM-style scheme for tuning the method’s hyper-parameters. Empirical results are shown on a toy grid-world domain for: imitation learning and system identification (learning of transition-dynamics) under both, a fixed random policy and a policy produced by a planner that takes into account the learned transition dynamics. The method is also evaluated on a “Queuing Network Modeling” task (121-dimensional state-space, 2 actions). The paper addresses a very important and timely problem: learning of higher-order statistical correlations in discrete MDPs (and related problems such as contextual bandits). While I think that the particular approach taken in the paper is very promising, the current manuscript has some shortcomings such that I currently vote and argue for major revision of the work. My main issues are (I) quality and extent of the experiments. (II) insufficient discussion of related literature and shortcomings/restrictions of the presented method. I appreciate that the authors show a number of different applications and I would be excited to see convincing results in favor of the proposed method and therefore want to encourage the authors to take the time to improve the manuscript. Detailed comments: 1) Experiments: the comparison shown in the experiments (“Dirichlet”) is ok as a naive baseline, but not enough as a serious competitor method. As a minimum the (block Gibbs) sampling version of the method should be included and it some discussion/results on strengths and weaknesses of the two variants (does the VI version run faster in terms of wall-clock time, is it more sample efficient, does it generalize better, …?). Given the small size of the toy domain, other (brute-force, or inefficient sampling-based) methods could potentially be included as well, but it would be OK to dismiss them by showing results on a larger-scale task where competitor methods can no longer be applied. Another competitor for comparison in Fig 1 would be the Dirichlet estimate with (a) copying the action-distribution from the nearest observed neighbour state or (b) taking the average over all observed neighbour states within a certain radius. 2) Larger-scale experiments. Why were there no experiments with larger state-action spaces and non-trivial dynamics included (at least grid-worlds with walls, and other non-trivial tiles)? Currently it is hard to judge whether this was simply due to a lack of time or because the method has severe scalability issues. Very convincing experiments would be e.g. on simple video-game domains, (which naturally have a low-cardinality discrete state- and action-space) - simulators for such experiments are publicly available and comparison against other approaches would be easier. 3) Literature: there is a considerable body of literature on (hierarchical) inference in latent-variable models that is barely mentioned. E.g. in the language domain, topic models are mentioned but dismissed as “not discussing the problem of decision-making”. Can some of these methods be straight-forwardly be applied to the tasks/domains shown in the paper - it seems so, since the model in the paper is not explicitly used for decision-making. Please correct me if I’m wrong and discuss this in greater detail. Hierarchical inference for discrete-variable models is also discussed in the literature on Bayesian deep (reinforcement) learning and hierarchical representation learning with deep networks - importantly, these models are also optimized via variational inference (ELBO maximization), however under different approximate distributions (the emphasis is on differentiability, rather than closed-form expressions). What are the advantages/disadvantages compared to the presented method (scalability, data-efficiency, …)? I am of course happy to also see non-deep-neural-network approaches, but this literature must be discussed in order to put the method into perspective. See e.g. [1] for lots of up-to-date pointers to literature. [1] https://duvenaud.github.io/learn-discrete/ 4) Shortcomings of the method and implications of the simplifications/approximations. Please discuss the implications of the mean-field approximation for the variational distributions, beyond simply stating the mathematical form. The same applies for restricting \theta to be a scale parameter (line: 165) - ideally compare empirically against no restrictions and doing the full matrix inversion numerically (particularly since the experiments are on small domains), or against using a low-rank matrix factorization. Finally, please discuss the implications of using a square-exponential kernel - would it for instance still be suitable in grid-worlds with walls, or other situations where simple Euclidean distance of states is not indicative of the “generalizability” of state-dependent action-distributions. Originality: Medium - the derivation of the VI scheme and the EM scheme is interesting and novel, but replacing a sampling-based ELBO optimization with a VI-based one is a rather straightforward idea. Quality: Low - while the derivations are well-presented and sufficient detail is given, the experimental section lacks comparison against important methods. Some ablation studies and sensitivity-analysis w.r.t. Hyper-parameters would have been nice and results regarding larger-scale applications are crucially required to judge the significance of the approach. The literature-discussion lacks important parts. Clarity: High - the paper is generally well written. The only important improvement is a qualitative/informal discussion of some of the restrictions/approximations, such as the mean-field approximation - though the mathematical statement is sufficient in principle, adding one, two sentences would not hurt. Significance: Currently low - the method could potentially be quite significant, but this needs to be shown with experiments that compare against other state-of-the-art methods and larger-scale experiments. It remains unclear whether the same results could have been achieved with the sampling-based approach and where the advantages of the VI approach lie.

[Author Response · NeurIPS 2019]

We would like to thank all reviewers for their constructive feedback and insightful comments. Here our answers:

**Reviewer #1**   As you stated correctly, the main contribution of this paper lies in the generality of the proposed inference framework, not in achieving higher scores than domain-specific algorithms. Nevertheless, we agree that a comparison with such algorithms will lead to a much stronger conclusion (see also Reviewer 3) and we will try our best to include additional baselines in the final version. Furthermore, we will make our code available with the camera-ready version.

We are aware that there exist approaches (like IRL) that are capable of generalizing behavior to unobserved situations by shifting the reasoning to the intentional level. However, please note that the very same logic also applies on that level: explicitly taking into account correlations between intentions (e.g. given in form of subgoals) can lead to better predictive models that outperform their correlation-agnostic counterparts while requiring less training data. To demonstrate this effect, we will include an additional IRL scenario in the camera-ready version, which we had to omit in the submitted version for space reasons. For details, please see comments to Reviewer 2 and preliminary results in the figure below.

**Reviewer #2**   We agree that the range of modeling scenarios to which our methodology can be applied is quite open. However, we consider this as a clear benefit as it emphasizes the generality of our approach, which enables many possible future developments. While planar navigation on a grid is a nice toy problem to demonstrate the working principle, the same methodology can be applied to arbitrary decision-making problems comprising discrete-valued elements. Moreover, apart from single-agent RL, we have many other potential use cases in mind, such as modeling correlations between agents in a multi-agent network and learning common structures across tasks (transfer learning).

In general, the proposed methodology can be beneficial in all scenarios that contain some form of structure (a wall in a grid world is just one particular example, just like the underlying grid layout itself and the associated translation-invariant transition dynamics). Often, these structures manifest themselves at different levels (e.g. similar intentions triggering similar actions) and applying our concept on one particular level can be superior to others, depending on the application. This effect is demonstrated by the situation shown in the figure below (preliminary results), which compares the PG imitation learning model from Section 4.1 that captures action correlations with one where the PG concept has been applied on the intentional level, capturing correlations between subgoals assigned to different states. In fact, the latter describes an additional use case of our framework, which we had to exclude from the paper due to space limitations but which will be described in detail in the camera-ready version (which is possible due to one additional page of content). The example shows a clear improvement for the subgoal-based variant, which generalizes the data on the intentional level (and not on the action level), yielding a significantly better reconstruction in unobserved regions of the state space. Note: the Dirichlet imitation learning model is omitted as its prediction is restricted to the visited states (compare Fig. 1b in the paper).

**Reviewer #3**   Many thanks for your feedback. Here are a few things that we would like to mention:

• We are eager to apply our approach to larger domains and real data. However, this requires further model approximations, which are out of the scope of this paper and which we leave for future work. Due the involved covariance matrix inversion, inference in our model scales cubically with the cardinality $C$ of the considered covariate space, currently preventing any large-scale experiments (similar to GP inference). Nevertheless, approximations similar to those used for GPs are possible, e.g. using local correlation models (compare inducing point method for GPs) or conjugate gradients.

• Please note that the derivations found in [12] (arXiv version) do neither include the variational principle for this particular problem type (instead, an LDA-like scenario is considered) nor the corresponding hyper-parameter optimization.

• We will discuss the suggested methods (Poupart and Texplore, see Reviewer 1) in the related work section of the camera-ready version and try to establish a performance comparison with our approach. However, note that a fair comparison might be difficult due to the conceptual differences of all approaches, i.e. the mentioned baseline do not capture the correlation structure explicitly and hyper-parameters need to be specified manually for each problem domain.

(a) expert policy & rewards    (b) data set & rewards    (c) policy reconstruction via **PG subgoal modeling**    (d) policy reconstruction via **PG imitation learning**    (e) policy reconstruction via **Dirichlet subgoal modeling**

Figure 1: Modeling action correlations vs. modeling subgoal correlations. Black squares indicate wall states. Red color indicates the per-state policy reconstruction in terms of the Hellinger distance to the expert policy.

[Meta-Review · NeurIPS 2019]

The paper develops a variational inference algorithm for modelling discrete state action MDPs. The model can be used to capture the correlations inherent between the states of an MDP. For doing so, Polya-Gamma auxiliary variables have been used, which has been proposed before. The contribution of the paper is the variational inference algorithm instead of using Block-based Gibbs sampling as in the original model. The paper attacks a very important problem, follows a nice idea and is well executed. We had 3 positive initial reviews. The reviews appreciated the contribution of the paper and the generality of the approach as well as the experiments. However, all reviews have been with a rather low confidence. We therefore acquired a 4th (emergency) review that was unfortunately not available for the rebuttal phase. The 4th review was very exhaustive but less positive. The main concerns of the reviewer were: - Missing comparisions to more convincing baselines. In particular, a comparison to the Gibbs sampling approach from [12] is crucial as this is the direct predecessor algorithm. - More complex experiments are needed to evaluate the scale-ability of the approach. I do agree with these points, but think they have been partially addressed in the rebuttal (the authors were not aware of R4). The authors stated results from a more complex problem domain (grid world with walls) and also explained that [12] does not contain a hyperparameter optimization procedure. I would still appreciate a comparison to [12]. but also given that the authors did not had a chance to reply to R4, I tend to still accept the paper as it contains very good ideas and the Poly Gamma model has been used for the first time for MDPs. However, the authors should extend the experimental section with the results given in the rebuttal.